# BavsNeRF: Batch Active View Selection for Neural Radiance Field Using Scene Uncertainty

You Li [1]    Rui Li [1,2*]    Shengjun Tang [2]    Yanjie Gu [1,2]

[1]Guangdong Laboratory of Artificial Intelligence and Digital Economy (SZ)
[2]Shenzhen University

liyougis@gmail.com    litun@gml.ac.cn    shengjuntang@szu.edu.cn    guyanjie@gml.ac.cn

## Abstract

*Active view selection is crucial for Neural Radiance Fields (NeRF) modeling in scenarios with limited number of posed images. Existing methods to select the views are either heuristic or computationally demanding. To address this, we propose a novel framework, BavsNeRF, to guide our view selection for NeRF modeling using scene uncertainty. We first establish an uncertainty estimation model of the entire scene based on an initial NeRF model. With this, we guide new perceptions by incorporating an batch active view selection policy, enabling the entire view selection procedure within a single iteration. In this way, the quality of novel view synthesis can be enhanced by incorporating images from selected viewpoints containing informative data. Experiments on both synthetic and real-world datasets demonstrate that the proposed method can identify informative new viewpoints, leading to more accurate scene reconstruction compared to baseline and state-of-the-art methods.*

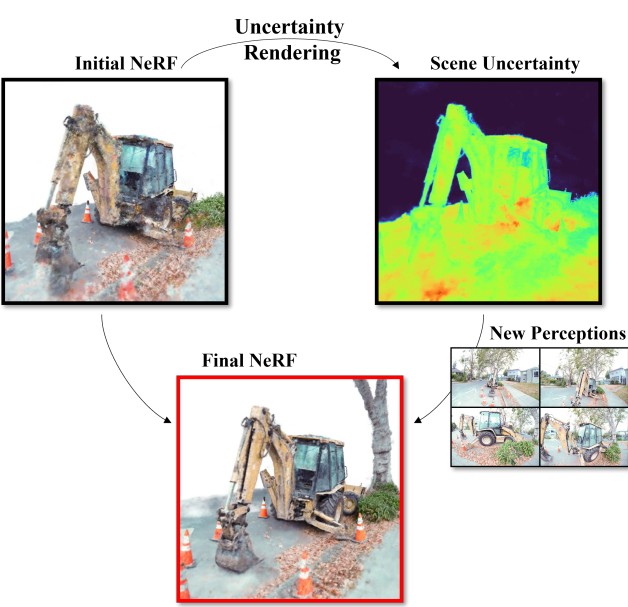

Figure 1. **Pipeline of the proposed method**.

## 1. Introduction

In recent years, the rapid advancement in technologies like autonomous driving, digital humans, and augmented reality has led to an increasing demand for high-quality real-world 3D models. Among these, the application of virtual humans has become a popular research topic, making the 3D reconstruction of the human body and face a fundamental yet essential task in computer vision. Dramatic improvements have been made on 3D human body reconstruction and face shape reconstruction in recent years. However, due to the lack of training datasets and the limitations of linear statistical models, 3D human body and facial texture reconstruction remains a challenging task. Additionally, conventional methods of 3D reconstruction, predominantly dependent on manual data gathering, find it challenging to meet this growing demand. As robotics technology evolves and matures, its numerous benefits such as efficiency, accuracy, flexibility, and security have played a major role in enhancing data acquisition for 3D reconstruction[22][19][6]. Employing smart robots for autonomous data acquisition has substantially broadened our proficiency and comprehension of real-world modeling[13][12][33].

Reconstruction results with smart robots heavily depend on the criteria for selecting new viewpoints for data collection. Traditional explicit scene representation methods for autonomous 3D reconstruction including point clouds, voxels and surfaces employ different strategies for selecting new viewpoints for data collection. In volume reconstruction tasks, [8], new viewpoints are selected by calculating the entropy of the voxels. Another method leverages a deep network based on point clouds to estimate the un-

---
*Corresponding Author.

certainty associated with different camera viewpoints. By considering the uncertainty, the system identifies the view that would most effectively reduce the existing model's 3D uncertainty[32]. Additionally, some researchers integrate Truncated Signed Distance Function (TSDF) mapping into an online planner for accurate 3D reconstruction while minimizing runtime, with their multi-robot systems[6].

Compared to traditional explicit 3D representations used for active view selection tasks, implicit representation methods like NeRF[15]have demonstrated great potential in several aspects, including precise geometric encoding, efficient memory consumption (adapting to scene size and complexity), predictive capabilities for filling unseen regions, and flexibility in training data volume. Recently, remarkable results have been achieved in 3D reconstruction, using offline images or online images captured with handheld cameras, through implicit neural representations[22]. However, quantifying uncertainty within NeRF models remains a nascent field. Some existing researches demand high computational resources[23][11] as they need to repeat traditional inefficient NeRF training[17]. Others rely on heuristic agents, which are not robust in certain scenarios[7][33]. Moreover, some other methods predict pixel uncertainty from nearby reference views through the pre-trained network, but they are inefficient for robots that require fast motion[9].

To address the aforementioned problems, a novel framework is proposed for selecting viewpoints for NeRF modeling using scene uncertainty. We achieve efficient NeRF modeling by incorporating a new uncertainty estimation algorithm and an efficient strategy for selecting target viewpoints. Figure 1 depicts the pipeline of our active view selection method. First, we train an initial NeRF model covering the overall range of the scene from randomly-selected views to obtain accurate training parameters. Second, we calculate the Hessian matrix of these training parameters based on the Bayesian posterior distribution, enabling pixel-level uncertainty estimation in the scene model through Fisher information. Additionally, we learn volume rendering techniques to output uncertainty estimation other than color as an additional channel in volume rendering, facilitating visualization of scene uncertainty. To overcome combinatorial explosion and local minima problems when selecting candidate views, we introduce a maximum minimum distance algorithm to constrain the relationship between candidate views and the initial training views. Furthermore, we limit the relationship between candidate views to avoid the aforementioned issues. We extensively evaluate our method on both synthetic and real-world datasets, comparing it with baseline and previous methods. Quantitative and qualitative results demonstrate the superiority of our approach. In summary, our contributions are as follows:

• We proposed a framework for the active view selection

of NeRF modeling that produces more accurate scene reconstruction results than state-of-the-art methods on both synthetic and real-world datasets.
• Our framework introduces a NeRF-based approach to quantify scene-level uncertainty for active view selection, avoiding the re-modeling of NeRF for every viewpoint estimation.
• Our framework introduces a maximum minimum distance algorithm to implement batch selection of viewpoints, thereby avoiding combinatorial explosion and the local minima problem.

## 2. RELATED WORK

### 2.1. Active View Selection

The active view selection task for robots equipped with RGB or RGB-D sensors is a popular research area[31]. The objective of this task for robots is to capture as much information as possible for 3D reconstruction by actively selecting new viewpoints in unknown scenes using their sensors, . According to the representation techniques for 3D models, previous methods are mainly divided into voxel-based methods[8][13], surface-based methods[6][28], and point cloud-based methods[32]. Initially, in unknown scenes, 3D reconstruction tasks iteratively select the next best viewpoint from a set of candidate views based on a known map state. Some studies[8] construct a probabilistic occupancy volume and calculate occlusion to compute information gain for selecting the next viewpoint, using voxels as the 3D representation. Wu et al.[28] utilize point clouds as their 3D shape representations and employ a poison field to obtain the current estimated confidence map, determining which parts of the object require further scanning. Although this method significantly improves the precision and quality of the 3D surface, its execution time is notably extended. All of the above methods require explicit 3D representations to store information about the current scene, leading to substantial memory consumption and restricting their scalability and representational potential. In contrast, our method merely involves two-dimensional images as input and utilizes implicit neural representation to determine the uncertainty of the entire scene. Therefore, it enhances the generalization capability of scene representation, reduces memory consumption, and finally improves the quality of the final 3D reconstruction.

### 2.2. Implicit Neural Representations

Implicit neural representation models 3D scenes as differentiable continuous neural networks[26]. For example, NeRF [15]learns density and radiance field values of the scene supervised by 2D images. Subsequent works have addressed NeRF's aliasing artifacts [2], Mller et al.[16], Garbin et al.[4] for faster training and inference, or Yu et al.[29] sug-

gest that training can be performed from a small number of views. As geometry for neural implicit representations can be represented continuously without discretisation, this representation technique does not rely on spatial resolution, and thus has a low memory footprint. Early works on implicit representations optimise the network, Park et al.[18] to regress a signed distance function (SDF) or occupancy function taking 3D coordinates as input. However, they all require the presence of 2D images as supervision and lacks the capability to deduce geometry shapes with an undetermined 3D state. Using implicit neural representations for active views selection is still a relatively new field, yet numerous studies have showcased its capabilities. [19][9][11].

## 2.3. Uncertainty in Neural Radiance Fields

Estimating uncertainty in Bayesian machine learning has been a popular research topic [10]. NeRF[15] implicitly represent a 3D scene through neural volume coding. By changing camera position and lighting in the scene, NeRF render the 3D scene with various uncertainties. Many recent works address the problem of NeRF uncertainty quantification. NeRF-W[14] directly learns the predicted RGB variance as a measurement of uncertainty for transient objects in the rendered scene. Shen et al.[21] proposes to learn to model all the probability distributions of possible radiance fields for the scene, treating radiance and densities as random variables, and approximating their posterior distributions after training using variational inference. Some recent works address the problem of time consumption in NeRF training by leveraging uncertainty to determine the best next view. Pan et al.[17] implement an active learning scheme which expands the existing training set with newly captured samples. Lee et al.[11] suggest using the entropy of density predictions computed along the rays as an uncertainty metric for NBVs selections. Other works[9][19] estimate uncertainty based on RGB images using neural networks. Goli et al.[5] introduce Bayesian posterior distributions to build volumetric uncertainty fields using Labras approximation. They select next best views by sampling points in space and rendering the uncertainty of corresponding points.

## 3. Approach

In this part, we present a novel batch active view selection framework named BavsNeRF, as illustrated in Figure 2. Our method is based on the posterior distribution of the initial NeRF model parameters and infers the uncertainty of the 3D scene by adding perturbations and using Laplace approximation. Viewpoints with high uncertainty values are considered in need of supplemented data collection. The rest of this section is organised as follows: Sec.3.1 describes the relevant formulas in the Neural Radiation Fields and the Laplace approximation, Sec.3.2 elucidates the overall framework process with the selection of the viewpoints, and

Sec.3.3 shows how to compute the 3D scene uncertainty and estimate the best candidate viewpoints.

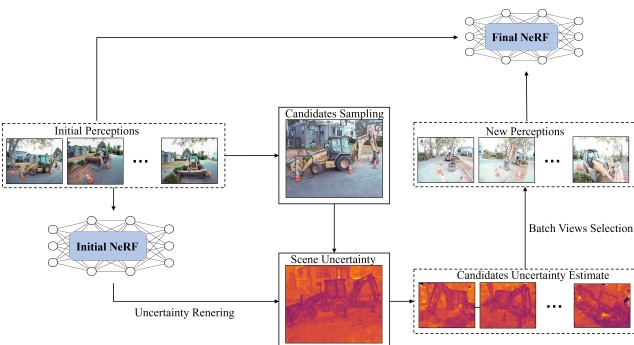

Figure 2. **BavsNeRF:** We first build the initial NeRF from initial perceptions. The NeRF model is then used to compute the scene uncertainty. After batch selecting views from candidate views, we can build the final NeRF model with better quality.

## 3.1. Preliminaries

NeRFs[15] models the continuous radiance field of a static scene by encoding the scene as a volumetric density ($\sigma$) and a colour $c = (r, g, b)$ through a multilayer perceptron (MLP) by parameterising the input 5D coordinates, including the coordinates of the 3D points contained in space $(x, y, z)$, and the direction$(\theta, \phi)$ of the rays emanating from the centre of camera projection $o$.

$$c_\phi(x, d), \tau_\phi(x) = MLP_\phi(x, d) \qquad (1)$$

where $x, d$ are both position-encoded 3D point coordinates and ray directions, and $\phi$ denotes the learnable model parameters in the MLP. The radiance color is only obtained from its own 3D coordinates as well as the viewing direction through the MLP with learnable parameters $\phi$. To perform novel view synthesis, using volume rendering techniques [24] along the camera rays $r = o_r + t \cdot d_r$ emitted from the camera center $o \in \mathbb{R}^3$ passing through a given pixel on the image plane, the color of that pixel can be represented as:

$$C(r) = \int_{t_n}^{t_f} T(t)\sigma(r(t))c(r(t), d)dt, \qquad (2)$$

where $T(t) = exp(-\int_{t_n}^{t} \sigma(r(s))ds$ represents the cumulative transmittance along the ray from $t_n$ to $t$, which is the probability of the ray propagating from $t_n$ to $t$ without intersecting any other particles. To handle the rendering process, NeRF employs hierarchical sampling to approximate the integral, representing it as a linear combination of discretized sample points:

$$C_\phi(r) = \sum_i exp(-\sum_{j<i} \tau_j\delta_j)(1 - exp(-\tau_i\delta_i))c_i, \qquad (3)$$

$\delta_i = t_{i+1} - t_i$ represents the distance between adjacent samples. Network parameters $\phi$ are optimized by minimizing the reconstruction loss, which is defined as the squared distance between the color predicted by the neural network and the actual color.

$$\phi^* = \arg\min_\phi \mathcal{L} \qquad (4)$$

$$\mathcal{L} = \|C_\phi(r) - C_{gt}^n(r)\|_2^2 \qquad (5)$$

Where $r$ is the sampled ray, $C_\phi(r)$ represents the model prediction based on the learned parameters $\phi$, and $C_{gt}^n(r)$ represents the ground truth value.

### 3.2. Uncertainty Estimation in Neural Rendering

Quantifying the uncertainty of neural networks typically involves using the Bayesian formula or its approximation for deep learning methods and studying the posterior distribution $p(\theta|D)$ conditioned on data $D$. In fact, we can approximate this posterior distribution using the Laplace approximation method. This approach is similar to what we previously used in simple linear regression problems with Bayesian methods, where we normalize the product of the likelihood Gaussian distribution and the prior Gaussian distribution to obtain the posterior Gaussian distribution. In Laplace's method, the goal is to find a Gaussian approximation $q(\theta)$ centered at the mode $\theta^*$ of the plurality of $p(\theta|D)$. such that $p'(\theta^*) = 0$. We approximate the probability distribution on the **M**-dimensional space $\theta$. The Gaussian distribution's logarithm is the quadratic function of the variables. Therefore, we consider the Taylor expansion of $\ln f(\theta)$ around the mode $\theta^*$.

$$\ln f(\theta) \simeq \ln f(\theta^*) + \frac{1}{2}(\theta - \theta^*)^\top \mathbf{H}(\theta^*)(\theta - \theta^*) \qquad (6)$$

where $\mathbf{M} \times \mathbf{M}$ the Hessian matrix $\mathbf{H}$ is

$$\mathbf{H} = -\nabla\nabla \ln f(\theta)|_{\theta=\theta^*} \qquad (7)$$

Taking the exponential of both sides of Eq.6 and then normalizing, we obtain the corresponding Gaussian likelihood,

$$\mathcal{N}(\theta|\theta^*, \mathbf{H}^{-1}) \qquad (8)$$

According to the [5], applying this framework directly to NeRF by identifying $\theta$ and $\phi$ is impractical. We also need to incorporate a deformation field $\mathbf{D}_{\theta_{(x)}}$ to introduce perturbations. As shown in Figure3, we consider introducing perturbations to compute uncertainty estimates. Furthermore, a regularized Gaussian prior $\theta \backsim \mathcal{N}(0, \lambda^{-1})$ should be imposed on the new parameters $\theta$. Subsequently, we compute the posterior distribution $p(\theta|I)$ by minimizing its negative log-likelihood $h(\theta)$, resulting in $\theta = 0$, which is the mode

of the distribution $p(\theta|I)$. According to Eq. 8, we obtain the Laplace approximation distribution $\theta \backsim \mathcal{N}(0, \Sigma)$ which

$$\Sigma = -\mathbf{H}(0)^{-1} \qquad (9)$$

where $\mathbf{H}$ is the Hessian matrix when the second order derivatives of $h(\theta)$ is zero. Computing these second-order derivatives is a computationally intensive task. Subsequent tasks approximate the Hessian matrix by using Fisher information, allowing us to approximate using only first derivatives.

$$F(\theta) = \mathbb{E}_{x \backsim \theta}\left[\frac{\partial^2 \log p(x|\theta)}{\partial\theta^2}\right] = -\mathbf{H}(\theta) \qquad (10)$$

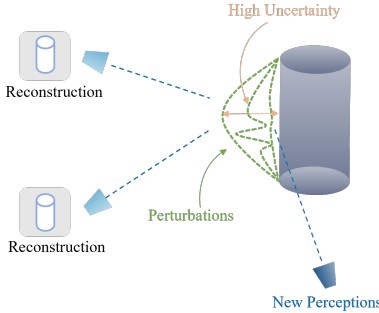

Figure 3. **Uncertainty estimated by introducing perturbation**: Uncertainty is positively correlated with size of introduced perturbations which will not affect the reconstruction quality. In the figure, left cameras represent the initial perceptions. After perturbations are introduced to the three-dimensional object, the reconstruction quality at the initial viewpoints remains unaffected. Regions with significant perturbations have high uncertainty values, thus are in need of new perceptions.

In the novel view synthesis problem, we use a set of random variables (r,y) corresponding to each ray r sampled from the training image dataset D and its corresponding true value y

$$F(\theta) = \mathbb{E}_{(r,y)}[4\epsilon_\theta(r)J_\theta(r)^\top J_\theta(r)] + 2\lambda I \qquad (11)$$

where $\epsilon_\theta(r)$ is the residual between the ray prediction and the true value.

$$\epsilon_\theta(r) = \|\tilde{C}_\theta(r) - C_{gt}(r)\|^2 \qquad (12)$$

Finally, based on Eq. 10 and approximating the expectation by sampling the rays R, we obtain the final expression for $\mathbf{H}$:

$$\mathbf{H} \approx -\frac{2}{R}\sum_r J_\theta(r)^\top J_\theta(r) - 2\lambda I \qquad (13)$$

From Eq. 13, we can see that the Hessian contains all the uncertainty information of the radiance field, originate from the camera parameters in both the training model and

the training data. It also simplifies the computation from the original second derivative Hessian matrix to the computation of the first derivative Jacobian determinant. Additionally, due to the sparsity of $\mathbf{H}(\theta)$, we can learn Ritter et al.[20] to approximate $\Sigma$ only by the diagonal $diag\mathbf{H}$ of $\mathbf{H}$. This approximation simplifies the calculation of the Hessian matrix when dealing with numerous rays. As illustrated in Figure 4, we incorporate uncertainty estimation into a NeRF model, which is rendered as an additional channel.

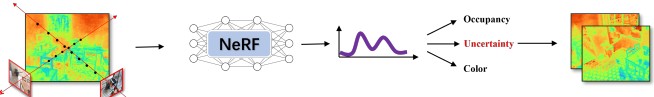

Figure 4. **Scene uncertainty estimation as an additional channel of NeRF.**

### 3.3. Uncertainty Guided View Planning

Our active view selection framework utilizes uncertainty estimated from image-based neural rendering to guide efficient data collection. Given a limited measurement budget, our uncertainty-guided approach can effectively find more informative images for better reconstruction of unknown scenes. For view planning, we uniformly randomly select candidate views from the scene-centric hemisphere space to evenly acquire corresponding viewpoints from various poses.

$$\phi = 2\pi u, \theta = \arccos\left(1 - v\right) \tag{14}$$

where $u, v \in [0.0, 1.0]$. However, if we calculate an uncertainty measure for each viewpoint without limiting its calculation scope, we will face the combinatorial explosion problem.

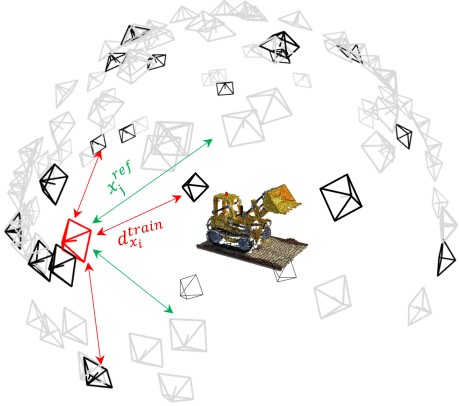

Figure 5. **Batch view selection policy:** We incorporate the maximum minimum distance algorithm to avoid combinatorial explosion and the local minima problem.

As shown in Figure 5, we first calculate the Euclidean

distance $d_{x_i}^{ref}$ from each candidate view $x_i^{ref}$ to each training view $y_j^{train}$ as well as the Euclidean distance $d_{x_i}^{ref}$ between candidate views. However, to prevent a drastic reduction in candidate viewpoints, we sort training views based on maximum and minimum distance, and limit the distance between candidate views. Ultimately, we select candidate viewpoints $D^{can}$ to estimate uncertainty, maximizing the restriction on the initial uncertainty selection and taking into account the correlation between candidate views. Moreover, to clarify the objective of our uncertainty measure, we also confine the uncertainty within a bounding box. This objective is achieved by setting the near and far planes in the rays.

For each candidate view, we estimate the uncertainty $U_i$ according to the calculation in Sec 3.2. Uncertainty is calculated based on the rays emitted from each pixel in the image, for a set of n candidate views $U_n \in H_r \times W_r \times scale$, where $H_r$ and $W_r$ are the pixel dimensions of the image, and scale is a factor to compress the image to prevent excessively large pixels. Uncertainty is computed with a simple function:

$$f_i = \frac{1}{W_r \times H_r \times scale}\|U_n\|_1 \tag{15}$$

High uncertainty views indicate the corresponding places need supplemental information. Candidate views with the k largest $f_i$ are appended as target place to collect data. Along with the initial collection of images, we build the final NeRF. Since our method can batch select viewpoints with high uncertainty, it can also be applied to perform path planning for large scenes.

## 4. Experiments

In this section, we present the experimental evaluations of our approach. In section 4.1 we focus on experiments on active view selection using publicly available real-world datasets. We present the results and compare our method quantitatively and qualitatively with previous methods. Then in section 4.2 we show additional experiments on uncertainty estimation, and finally in section 4.3 we present the algorithmic ablations. The experimental results support our three claims: (1) Our strategy for active view selection is reasonable and practically meaningful. (2) The uncertainty model we developed, based on neural rendering, plays a crucial role in selecting perception views and is applicable in large scenes; (3) Estimating uncertainty offline is beneficial for many applications.

### 4.1. Active View Selection

The feasibility of our active view selection method is validated by assessing if the required perception views can be identified through expected information gains. We first present our dataset as well as experimental settings,

and then compare our method qualitatively and quantitatively with baseline randomised methods and state-of-the-art methods.

#### 4.1.1 Datasets

Our approach was validated on three datasets, including the original real-world dataset provided by nerfstudio [25], the real-world Mip-NeRF360 [3] dataset, and the border-less synthesis Blender dataset [15]. The nerfstudio dataset contains contains 10 scenes: 4 mobile phone photos taken with a pinhole lens and 6 mirrorless camera photos taken with a fisheye lens. It serves to assess the proposed method in real-world scenes, utilizing nerfstudio'sstandard "nerfacto" training setup. The Mip-NeRF 360 dataset contains 9 different scenes from real-world 360° captures, employing an identical training setup as used in the nerfstudio dataset. The Blender dataset of the unbounded synthetic scene, which contains 8 complex geometric shapes, is divided into 100 views as the training set and 200 views as the test set. We select the initial images from the 100 training views, and the rest as the candidate views. Due to the poor training results of the "nerfacto" method on unbounded synthetic scenes, the default training setup in the Mip-NeRF 360 dataset is employed.

Table 1. **Quantitative results on Blender Dataset.** We select 20 observations of objects from Blender Dataset. ActiveNeRF*: Performance using ActiveNeRF's active view selection strategy and our 3D reconstruction algorithm. Ours: View selection using our uncertainty estimation and without using the active view batch selection policy. Ours*: View selection using our uncertainty estimation and the active view batch selection policy. Our method achieves higher reconstruction quality compared to other methods.

| Method | PSNR↑ | SSIM↑ | LPIPS↓ |
|---|---|---|---|
| ActiveNeRF | 26.240 | 0.8560 | 0.1240 |
| Random | 27.138 | 0.9206 | 0.0652 |
| AcitveNeRF* | 27.346 | 0.9163 | 0.0743 |
| Ours | 27.997 | 0.9306 | 0.0696 |
| Ours* | **29.023** | **0.9328** | **0.0405** |

#### 4.1.2 Metrics

To evaluate the performance of view planning, test views are rendered with the test dataset and ground truth 3D model are reconstructed via neural radiation fields. The rendering quality is measured by Peak Signal-to-Noise Ratio (PSNR) and Structural Similarity Index Metric (SSIM)[27], in addition to LPIPS[34], which reflects human perception more accurately.

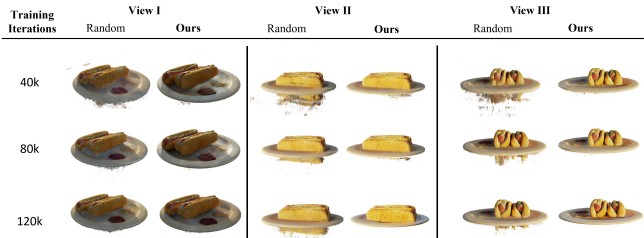

Figure 6. **Comparison of our method with random baseline.** Our experimental configuration is the same as that of AcitveNeRF. The initial training set has 4 images, and we capture 4 new perceptions every 40K iterations. The results show that our method outperforms the random baseline approach on novel view synthesis tasks.

Table 2. **Quantitative results on NeRFstudio Dataset.** We selected 200 images from the dataset as the candidate view set and the rest as the test set. We first select 50 random views as the initial training set and batch select 10 new views for the final 3D reconstruction. Origin denotes the quality of the initial 3D reconstruction. Our method exhibits significant improvement in quality compared to the initial NeRF, and outperforms the random baseline method.

| Method | PSNR↑ | SSIM↑ | LPIPS↓ |
|---|---|---|---|
| Origin | 16.5965 | 0.6184 | 0.4783 |
| Random | 17.1705 | 0.6283 | 0.3954 |
| Ours | **19.7318** | **0.7951** | **0.2953** |

Table 3. **The uncertainties computed on LF Dataset.** A lower AUSE[1] value means higher uncertainty estimate. The uncertainties estimated with our method on the Light Field dataset are significantly more accurate in calculating the real NeRF depth error than the previous state-of-the-art method CF-NeRF

| Method | Torch↓ | Africa↓ | Statue↓ | Basket↓ | Average↓ |
|---|---|---|---|---|---|
| CF-NeRF | 0.88 | 0.34 | 0.47 | **0.26** | 0.49 |
| Ours | **0.23** | **0.28** | **0.19** | 0.29 | **0.24** |

Table 4. **Ablation on candidates acquisition.** Our* indicates that we employ the batch view select policy, while our indicates not. The experimental results show that our* accelerates the process of view selection and acquires a higher quality 3D reconstruction.

| | Time↓ | Candidates | PSNR↑ | SSIM↑ | LPIPS↓ |
|---|---|---|---|---|---|
| Random | - | - | 17.1705 | 0.6283 | 0.3954 |
| Ours | 1408s | 250 | 17.2311 | 0.6101 | 0.4125 |
| Ours* | 197s | 43 | **19.7318** | **0.7951** | **0.2953** |

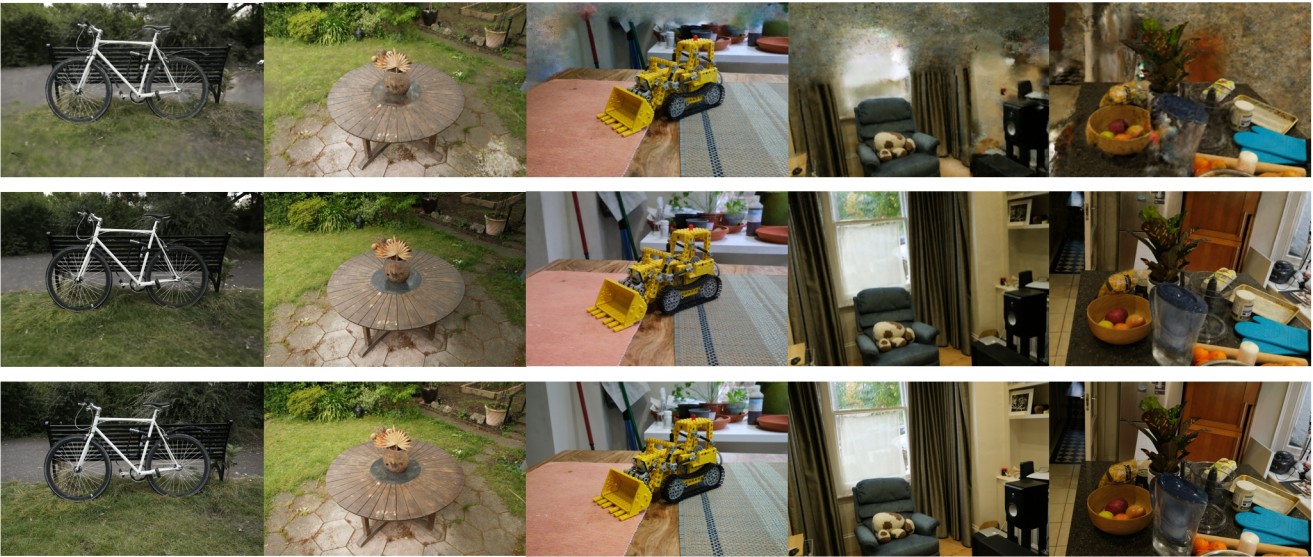

Figure 7. **Qualitative Study of our method on the Mip-NeRF 360 Dataset.** All models in this figure are implemented using the nerfacto method. From the top to bottom are results from ActiveNeRF, our method, and the ground truth.

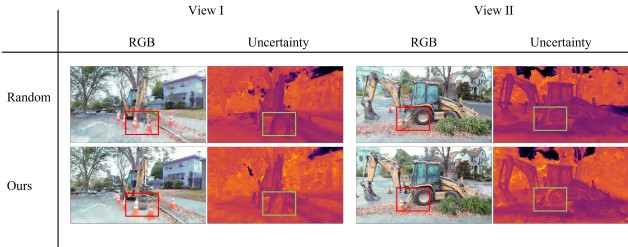

Figure 8. **Comparison of Ours method with random baseline method on Uncertainty.** From the figure we can see the color image and scene uncertainty estimation from different viewpoints using our method and the random baseline method.

### 4.1.3 Result

We validate the performance of our proposed framework in novel view synthesis tasks and compare it qualitatively and quantitatively with the state-of-the-art methods ActiveNeRF and the random baseline method. To ensure equitable comparison, both the reconstruction methods and other training setups employ identical setups, with the exception of the varied policies utilized by the algorithms. On the Blender dataset, we adopt the same experimental configurations as ActiveNeRF. With 4 images in the initial training set, we capture 4 new views every 40K iterations. The experimental results are shown in Table.1 and Figure.6, which shows that our method outperforms the rest.

We perform qualitative experiments on the Mip-NeRF 360 dataset and compared with state-of-the-art methods. The initial training set comprised 10 images. Every 3k iterations, 10 new perceptions are appended until 30k itera-

tions are completed.The results are shown in Figure7. The results demonstrate that our method outperforms the state-of-the-art. Qualitative and quantitative experiments are also conducted on the nerfstudio dataset.We also conducted qualitative and quantitative experiments on the NeRFstudio dataset. We selected 50 images as the initial training set and divided the remaining dataset into candidate view set and test view set. In this experiment, we trained the initial NeRF model using the initial training set and selected 10 new perceptions in batches to train the final NeRF model.The results are shown in Table.2 and Figure.8.The experimental results show that adding 10 new perceptions significantly improves the quality of 3D reconstruction, and outperforms the baseline method.

## 4.2. Evaluation of Uncertainty Estimation

As discussed in Sec 3.3, our model can also calculate the pixel-level uncertainty of training views. To evaluate the structural similarity following previous approaches, we introduce the Area Under the Sparsification Error curve (AUSE) metric to estimate the quality of the uncertainty estimates, i.e., how much they coincide with the true errors. A lower AUSE[1] means higher uncertainty estimates quality. We evaluate our method on the Light Field (LF) Dataset[30] and compare it with CF-NeRF, the previous state-of-the-art method. The results are consistent across Table. Ours uncertainty estimation method shows significant improvement in correlation with depth error compared to CF-NeRF. The results are consistent across Table.3.

## 4.3. Algorithmic ablations

To test the effectiveness of the proposed view candidates selection method, ablation studies are conducted in terms of time consumption and quality of 3D reconstruction with and without our maximum-minimum distance algorithm, using the same uncertainty model. We experiment on the dozer scene from the nerfstudio dataset, where we use 300 of images as candidate views and selecte 50 of them as initial training views. The algorithm actively batch select 10 new views for the final 3D reconstruction. We use the time metrics to represent the time it takes us to batch select new perception views from candidate views. And the candidates metric indicates the number of candidate views we need to compute the uncertainty metric when selecting a new perception view from the candidate view sets. Table.4 illustrate that incorporating a distance limitation algorithm enhances the efficiency of the active view selection algorithm and enriches the information from the selected new perception views.

## 5. conclusions

We introduce BavsNeRF, an active view selection framework utilizing NeRF to estimate scene uncertainty and guide new viewpoints selection for perceptions. By estimating uncertainty, we identify informative viewpoints in the scene, leading to high-quality scene reconstructions. This work can easily be applied to guide reconstruction of buildings with drones or digital human reconstruction with cameras. However, the limitation lies in the long training time for NeRF when optimizing network parameters. To address this, we may utilize faster rendering technique like 3D Gaussian Splatting to model the scene and estimate uncertainty as attributes of Gaussian elements, in our future works. Additionally, to extend our framework to more complex and cluttered environments, we plan to incorporate geometric uncertainty estimation into planning with unconstrained action spaces.

## Acknowledgements

This work was supported by the Natural Science Foundation of Guangdong Province of China(No.2023A1515010717) and the National Key Research and Development Program of China(No.2022YFC3800602).

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
