# OpenReview forum: "BavsNeRF: Batch Active View Selection for Neural Radiance Field Using Scene Uncertainty"
_thecvf.com/CVPR/2024/Workshop/POETS — CVPR 2024 Workshop POETS Poster_

### Official Review · Reviewer_q4oT · 2024-05-04
**Review of BavsNeRF**

**Rating:** 5
**Confidence:** 4

**Review:**

Summary:

This paper aims to propose a framework for view selection in the NeRF optimization. The key motivation is to quantify the uncertainty in NeRF training. The author uses the Fisher information during training to produce an uncertainty channel and leverage it to design a view-selection policy. The proposed framework outperforms the previous approach ActiveNeRF by a clean margin.

Strength:
1. The paper is well-motivated and the proposed solution is reasonable.

Weakness:
1. Lack of novelty. There is one paper FisherNeRF [a] shared similar motivation and designs as BavsNeRF. Both of these papers use Fisher information to quantify the uncertainty. The key formulations of these two papers are kind of similar.
2. The acknowledgment should not appear in the submission.
3. The writing could be further improved in the format (e.g, line 333, 3.2 -> sec 3.2), and caption of section (e.g. Conclusions in Sec5)

[a] FisherRF: Active View Selection and Uncertainty Quantification for Radiance Fields using Fisher Information

---

### Official Review · Reviewer_pULX · 2024-05-05

**Rating:** 7
**Confidence:** 4

**Review:**

This paper proposes an active view selection method for NeRF. The authors propose to estimate uncertainty actively based on Hessian matrix of training parameters and then select next training views based on estimated uncertainty. Basically this is a good paper and the proposed method achieves good performance, while I have several minor concerns:

1. The paper requires further proof reading to avoid small typos. For example: (1) two “.” in “with their multi-robot systems.[6].” in Line 048, Page 1; (2) Mip-NeRF dataset should be Mip-NeRF 360 dataset in line 388, Page 6.
2. The authors claim that the proposed uncertainty estimation method is more efficient than previous method (line 064-067), while there is no corresponding experiments to prove that.
3. It would be better to provide qualitative comparison with CF-NeRF.

---

### Official Review · Reviewer_A9YU · 2024-05-10
**The proposed approach is sound, but its effectiveness should be further validated**

**Rating:** 7
**Confidence:** 4

**Review:**

1. With the introduction of 3D Gaussians, the significance of conducting Active View Selection for NeRF may not be as substantial.
2. Can this set of strategies be transferred or adapted to work with 3D Gaussians?
3. How robust is the method? Is it capable of functioning effectively on datasets such as Tanks & Temples or aerial photography data?

---

### Meta-Review · Program_Chairs · 2024-05-14

**Recommendation:** Accept (Poster)
**Confidence:** 5

**Metareview:**

The paper introduces an active view selection method for NeRF, employing uncertainty estimation based on the Hessian matrix to guide training view choices, which is generally well-received. Strengths include a clear motivation and reasonable methodological advances, though concerns are raised regarding novelty and proofreading errors. Also, the discussion of multi-view synthesis with virtual humans should be clearly included. One critical issue inferred by the review is that authors include an acknowledgment in the under-review paper, which is a severe problem and should be avoided in future submissions.
In conclusion, this paper will be accepted. However, the weaknesses should be carefully considered and reflected in the final version.

---

### Decision · Program_Chairs · 2024-05-14

Accept (Poster)